# Bragg Peak Localization with Piezoelectric Sensors for Proton Therapy Treatment [note 1]

**DOI:** 10.3390/s20102987

**Published:** 2020-05-25

**Authors:** Jorge Otero, Ivan Felis, Alicia Herrero, José A. Merchán, Miguel Ardid

**Affiliations:** 1Institut d’Investigació per a la Gestió Integrada de les Zones Costaneres (IGIC), Universitat Politècnica de València (UPV), Gandia, 46730 València, Spain; mardid@fis.upv.es; 2Centro Tecnológico Naval y del Mar (CTN), Fuente Álamo, 30320 Murcia, Spain; ivanfelis@ctnaval.com; 3Institut de Matemàtica Multidisciplinar, Universitat Politècnica de València (UPV), 46022 València, Spain; aherrero@mat.upv.es; 4Grupo de Física Nuclear Aplicada y Simulación, Universidad Pedagógica y Tecnológica de Colombia (UPTC), 150003 Tunja, Colombia; jose.diaz@uptc.edu.co

**Keywords:** piezoelectric sensors, hadrontherapy, monitoring Bragg peak, FEM method, Monte Carlo simulations

## Abstract

A full chain simulation of the acoustic hadrontherapy monitoring for brain tumours is presented in this work. For the study, a proton beam of 100 MeV is considered. In the first stage, Geant4 is used to simulate the energy deposition and to study the behaviour of the Bragg peak. The energy deposition in the medium produces local heating that can be considered instantaneous with respect to the hydrodynamic time scale producing a sound pressure wave. The resulting thermoacoustic signal has been subsequently obtained by solving the thermoacoustic equation. The acoustic propagation has been simulated by FEM methods in the brain and the skull, where a set of piezoelectric sensors are placed. Last, the final received signals in the sensors have been processed in order to reconstruct the position of the thermal source and, thus, to determine the feasibility and accuracy of acoustic beam monitoring in hadrontherapy.

## 1. Introduction

The use of heavy particles as protons in the treatment against solid tumours has been considered a preferable treatment compared with the classical radiation treatment due to a better control of the localization of the energy deposition produced with the proton beam. This technique can be used in accessible in areas where conventional photon radiation is not possible because of the high damage in near healthy tissue [1,2,3,4]. The rapid advance in the field is seen in the number of accelerators in new centres of proton therapy [5,6,7]. The main advantage of protons over photons is the possibility of setting the dose in the tumour efficiently, minimizing the damage to tissue out of the region near the Bragg peak region. The location of the Bragg peak profile depends mainly on the initial energy of the beam and its width [8]. Among others, this deposition is particularly important in clinical conditions such as melanoma of the eye or tumours in brain tissue, Ependyoma, Germinoma, Craniopharyngioma, inter alia, where the possibility of damage to nearby tissue increases due to the size of the tumour tissue and the critical location [9]. In all cases, monitoring and control systems are carried out before and during each radiation session. Different alternatives for the monitoring process have been proposed based on physical processes produced in the irradiated medium such as prompt-gammas, charged particles, ββ-emitter or PET-imaging [10]. This article proposes an alternative monitoring method based on the pressure signal produced by the proton beam; whose amplitude is evaluated from a set of piezoelectric sensors [11]. In particular, it is based on simulation studies of the absorbed dose distribution for a set of energies using Monte Carlo methods that take into account the verification of the simulations with experimental data and the acoustic characterization of the piezoelectric transducers [12,13,14,15,16,17,18,19,20,21]. Thus, the thermoacoustic model calculates the close pressure spatial distribution which will be the input parameters for the Finite Element Method (FEM) that solves the acoustic propagation through the brain and skull, and the behaviour of the piezoelectric effect in a PZT material of the monitoring sensors.

In this process, there are additional considerations regarding the adaptive layers such as meninges among brain tissue and skull, the thickness of the skull and accelerator characteristics as protons per pulse, time profile and the width of the spatial profile inter alia. Firstly, the influence of the human brain tissue as meninges has been evaluated with respect to its acoustic behaviour in the propagation event [22,23,24]. The presence of thin layers of about a few microns is very important for the acoustic propagation given its acoustic properties and impedance. The detailed structure has not been taken into account in the simulation in order to reduce the computational process, however a time correction according to its acoustic impedance has been considered. Secondly, to evaluate the best time correction because of the difference of thickness throughout the skull, studies about acoustic and mechanical properties of the skull were reviewed to choose a mean value where the piezoelectric sensors were fixed [25,26].

Regarding the issue of temporal and spatial profile used in this study, in clinical applications, the value of the protons per pulse has increased in the last years due to the advances in the development of accelerators and software control [27,28,29,30,31]. This has allowed the application of fewer sessions in practical cases and faster results, whereby it is expected an increase in the final pressure signal on the surface of the piezoelectric transducer. However, in order to guarantee an acoustic signal, temporal and spatial profile and proton per pulse values have been taken according to studies on the acoustic signal detection threshold in accelerators for medical applications [32]. Hence, in clinical applications the values for protons per pulse are higher, whereby, a pressure signal is always guaranteed on the surface of the skull [33,34].

## 2. Model and Methods

### 2.1. Energy Deposition: Monte Carlo Simulations

In order to calculate the energy deposited in the water due to the Bragg peak, Monte Carlo methods have been applied using the Geant4 hadron therapy simulation package. The interface provides a complete set of physics processes to model the behaviour of particles and materials used to model complicated processes physics processes in complex geometries thanks to the fact that it is possible to modify the system with minimal effort [18]. The model features a monoenergetic proton beam using the Electromagnetic Quark Gluon String model with Binary Cascade for primary protons (QGSP BIC EMY) as Physics List [27] in energies among 80 MeV to 150 MeV. The absorbent material is G4Water with density 1003 kg/m^3^ with a number of simulations about 10^6^ histories. The detector volume consists of a solid G4Box with a voxel size of 1 mm. To evaluate the boundary conditions of bone tissue, the Anthropomorphic model has been used within the Geant4 libraries. The models are simulated in the volume of water including the bone structure. Generally, in medical physics applications, the results of Geant4 are evaluated in the dose deposit per unit area, however in this study the results are evaluated according to the energy of the beam deposited in the volume tissue. Thus, the behaviour of the Bragg peak will be used taking into account its energy part in the thermoacoustic model described in the following section.

### 2.2. Thermoacoustic

According to the thermoacoustic model [10], the deposition of energy from particles that pass through a liquid produces a local heating in the medium that can be considered instantaneous with respect to the hydrodynamic time scale. Due to the temperature change, the medium expands or contracts according to its coefficient of volumetric expansion α. The accelerated movement of the heated medium forms a pressure pulse that propagates through the fluid. The model is described through Equation (1) with the particular solution given by Equation (2).
(1)$${\bar{\nabla }}^{2}p\left(\vec{r},t\right)=-\frac{1}{c_{s}^{2}}\cdot \frac{\partial^{2}p\left(\vec{r},t\right)}{\partial t^{2}}=-\frac{\alpha }{C_{p}}\cdot \frac{\partial^{2}\epsilon \left(\vec{r},t\right)}{\partial t^{2}}$$
(2)$$p\left(\vec{r},t\right)=\frac{1}{4\pi }\frac{\alpha }{C_{p}}\int_{V}^{}\frac{dV^{\prime }}{\left|\vec{r}-{\vec{r}}^{\prime }\right|}\cdot \frac{\partial^{2}}{\partial t^{2}}\epsilon \left({\vec{r}}^{\prime },t-\frac{\left|\vec{r}-{\vec{r}}^{\prime }\right|}{c_{s}}\right)$$
where $p\left(\vec{r},t\right)$ is the pressure at a time t and spatial position $\vec{r}$, $c_{s}$ represents the speed of sound in the medium, $C_{p}$ is the specific heat, $\epsilon \left(\vec{r},t\right)$ the energy density deposited in the medium and α the coefficient of thermal expansion. When a pulse of protons radiates in a homogeneous volume, it corresponds to a pressure source proportional to the energy deposition $\epsilon \left(\vec{r},t\right)$. Each differential volume element of the irradiated source pressure emits a micro bipolar pressure wave. The pressure measured in the sensor is the sum of the micro-pressure waves emitted by each differential element of the volume [35]. Doubling the number of particles in a given pulse will double the resulting emitted pressure amplitude. In contrast to photoacoustic, the unique dose deposited by pencil-beam proton radiation creates two macroscopic pressure waves in a homogeneous medium: the pre-Bragg peak volume emits a cylindrical wave that propagates lateral to the proton beam direction while the Bragg peak emits an approximately spherical wave [20].

Due to the constant speed of sound throughout the medium, from the perspective of the sensor, the pressure that arrives after a certain time *t* is related to the pressure waves emitted in a common radius. As shown in Equation (1), the derivative of these pressure waves translates a bipolar behaviour that is finally the signal received by the sensor.

### 2.3. Acoustic Propagation: FEM Simulation

To deal with the propagation of the pressure calculated at a certain distance from the Bragg peak, a FEM model has been implemented in COMSOL Multiphysics that combines the Pressure Acoustics, Transient and Solid Mechanics interfaces to connect the acoustics pressure variations in the fluid domain with the structural deformation in the solid domain. Basically, the default sound pressure function models harmonic sound waves in the water domain using Equation (3) for sound pressure:(3)$$\frac{1}{\rho_{0}c_{s}^{2}}\frac{\partial^{2}p}{\partial t^{2}}+\nabla \cdot \left(-\frac{1}{\rho }\left(\nabla p-q_{d}\right)\right)=Q_{m}$$
where $\rho_{0}$ represents the density and $Q_{m}$ is the monopole domain source that corresponds to a mass source on the right-hand side of the continuity equation. The dipole domain source $q_{d}$, corresponds to a domain force source. The combination $\rho_{0}c_{s}^{2}$ is called the adiabatic bulk modulus, commonly denoted β. The bulk modulus is the inverse of the adiabatic compressibility coefficient $\beta =1/K_{s}=1/\rho_{0}c_{s}^{2}$ [36]. The pressure generated by the thermoacoustic model is distributed in a sphere within the skull corresponding to the range of the Bragg peak deposition. Figure 1 shows the geometric model and the positions of the sensors. The pressure value has been propagated to a spherical volume of 2 mm in diameter, which corresponds approximately to the Bragg peak for 100 MeV as shown in Figure 1b.

### 2.4. Piezoelectric Optimization

The received signal is related to the acceleration on the surface of the skull. This signal can be converted to pressure, and thus, to have a voltage reference due to the RVR of the sensor attached to an adaptation layer. The signal received by the sensors corresponds to an acceleration analysis using the FEM method included in the solid mechanic module. In this case, the matrices of mechanical and electrical behaviour of a piezoelectric material PIC255 has been implemented, where a differential displacement is produced on one of its surfaces due to the acceleration on the surface of the skull. In addition to the numerical method, the impedance and admittance results are compared with experimental measurements for a PIC255 disc ceramic with 25 mm in diameter (d) and 2 mm thickness (th). The measurement method is based on the electrical response using an impedance analyser as described in ref. [37]. Figure 2 shows the ceramics used and the mesh geometry to validate the numerical model.

Previous studies on the optimization of the geometry of a piezoelectric ceramic based on the frequency response of the expected signal have shown that it is possible to determine a geometry that improves the Receiving Voltage Response (RVR) sensitivity for the application with proton beams and knowing the frequency and bandwidth characteristics of the pressure pulse obtained [37]. Figure 3 shows a general outline of the optimization process.

As first step for the design of the piezoelectric device we optimise the radius and thickness parameters that influence the radial vibrational mode. The diameter and thickness range are set between 5 mm and 40 mm. Then, the impedance modulus is taken from the numerical model for each frequency. In the first and second radial mode, the minimum value in impedance represents the resonance frequency and the maximum value of the impedance the anti-resonance frequency, respectively. In step 3 the result of the pressure signal spectrum obtained with the thermoacoustic model is shown based on input parameters such as energy, beam emission time, beam width and the number of protons per pulse. This frequency response will be the parameter to be optimized in the design of piezoelectric ceramics. Thus, in the fourth step, the behaviour of the electromechanical coupling coefficient for low frequency vibration modes has been evaluated. This coefficient represents a relationship between the resonance frequency and the anti-resonance of the vibration mode evaluated $k=\sqrt{1-{\left(f_{r}/f_{a}\right)}^{2}}$, whereby it is possible to study the behaviour of this coefficient for the first two vibration modes that represent a bandwidth depending on the geometry of the piezoelectric ceramic. As a result, the highest values of the $k_{1}/k_{2}$ ratio have been taken, for the geometry that best fits the desired bandwidth.

### 2.5. Acoustic Source Localization

For the reconstruction of the position of the Bragg Peak, it has been used an acoustic source localization method based on the measurement of the DTOA (different time of arrival) of the acoustic signal between each pair of hydrophones. This analysis has been implemented following the previous studies [38] where it is explained the processing techniques to detect the arrival time, details of the location algorithm and the principles of resolution method of the resulting nonlinear system equation. However, in this case, it has been used a more robust version of Newton-Rapson method that ensures the convergence of the method [39].

## 3. Results

### 3.1. Energy Deposition

As a result of the interaction of protons with matter following the behaviour of the Bragg peak, it is possible to determine the energy deposited in the medium for a 100 MeV proton beam. For this, the deposition in the medium is evaluated directly and with a 1 cm bone layer that simulates the skull following the geometric models of the Geant4 libraries [40,41]. Thus, Figure 4 shows the Bragg profile as a function of the range in water, with the bone layer described and the distribution of this energy in a plane.

### 3.2. Thermoacoustic Signal

With the results of energy deposition and defining a time profile of 10 μs and a total of 5 · 10^6^ protons per pulse, it is possible to determine the pressure at one point in the space using the thermoacoustic model [38]. Since most of the energy is deposited in a volume that has a depth of approximately 2 mm, it is necessary to propagate the pressure from the simulation of different points in the space on the beam emission axis. For this, the received pressure for a 5 mm to 40 mm sensor in 5 mm steps has been simulated. The expression $P=P_{0}/d\cdot e^{-\alpha^{\prime }d}$ was used to calculate the propagated pressure, where $P_{0}$ is a known reference pressure at 5 mm, d the distance between the source and the sensor and $\alpha^{\prime }$ the water absorption coefficient (0.95 Np/m for 300 kHz). Figure 5 shows the pressure signal received by a sensor located 20 mm from the Bragg peak. The maximum pressure values for different distances assuming a pressure of 2.92 Pa at a distance of 2 mm is also shown.

Once the pressure behaviour has been simulated and established over time, the pressure value at a distance of 2 mm is set in the FEM model for wave propagation at a volume that corresponds to the Bragg maximum peak shown in Figure 4b.

### 3.3. Acoustic Propagation

The Time of Arrival (TOA) is influenced by the presence of different layers between the source and the sensor with different sound speed and acoustic impedances. According to the angle of incidence of the pressure pulse, the angle of the transmitted wave changes due to the appearance of longitudinal and shear waves through the solid skull. This behaviour can be evaluated according to a transmission model from which can be obtained the power transmission (T) and reception (R) in longitudinal (p) and shear (s) coefficients of the cerebrospinal fluid-skull interface [42]. Figure 6 shows the transmission angle $\theta_{t}$ from an angle of incidence $\theta_{i}$ on the skull for a longitudinal and shear wave in a solid-liquid interface between the cerebrospinal fluid of the skull and the parietal bone [42].

Once the acoustic pressure was calculated using the thermoacoustic model and obtained the propagated pressure to a source whose volume coincides with the energy deposition, the propagation in the brain is performed using a FEM model. For this, a uniform density has been taken in the region inside the skull with a value of 1003 kg/m^3^ that corresponds approximately to the density of water, due to the brain tissue and the cerebrospinal fluid have a similar density [43]. Figure 7 shows an image of the propagation in a horizontal plane in an instant of time t = 53 μs. In addition, the signal pressure signal received by the bone inside the skull and the acceleration on the outside surface of the skull are shown.

Thus, it is possible to determine the distance d_t_ given the known angle $\theta_{i}$. In this particular case, a correction of 1.08 μs in the TOA times has been applied in the sensor set.

### 3.4. Piezoelectric Optimization

To study the behaviour of a piezoelectric ceramic in the width of a specific bandwidth, it is necessary to evaluate a central frequency that is a product of the pressure profile shown in Figure 5a. Whereby and as a result of the impedance and admittance simulations for the radius and thickness values set out in Section 2.4, Figure 8a shows the values of the resonance and anti-resonance frequency as a function of the geometry changes in the ceramics for the first vibration mode. From the resonance and anti-resonance values for the first two vibration modes, Figure 8b shows the ratio of the electromechanical coupling coefficient $k_{1}/k_{2}$. In the specific case of the thermoacoustic result seen in Section 3.2, the spectrum of the pressure signal in Figure 5a has a central frequency of 110 kHz. Keeping it in mind, Figure 8a also shows a horizontal plane representing the central frequency of the pressure pulse. The area that intercepts this plane with the resonance and anti-resonance frequencies can be transferred to Figure 8b of the ratio $k_{1}/k_{2}$. In this figure, the highest value of the ratio $k_{1}/k_{2}$ has been chosen.

According to the electromechanical coupling coefficient ratio, the best option to optimize the sensitivity for 110 kHz is using a diameter (d = 15 mm) and a thickness (th = 9 mm) where the relationship between $k_{1}/k_{2}$ is maximum. With this in mind, the voltage value at terminals of this ceramic is shown in Figure 9.

The optimized geometry, in addition to complying with the bandwidth studied, presents an improvement in the *Receiving Voltage Response* (RVR). To validate the simulated results, the RVR for the PIC 255 ceramic shown in Figure 2a has been measured in the laboratory and its electrical behaviour has been simulated. With this, Figure 10 shows the results for the ceramics measured in the laboratory, d (25 mm), th (2 mm), along with its simulation and the simulation for optimized geometry d (15 mm), th (9 mm) for a central frequency of 110 kHz.

In this figure, the experimental and simulated RVR for the studied ceramic is shown, where the RVR corresponds to −210 dB re V/μPa in frequency. In addition, the new geometrical result presents a substantial improvement at the same frequency with a value of −185 dB re V/μPa.

### 3.5. Acoustic Source Localization

In a first approach, the signal arrival times are obtained by cross correlation between the emitted signal (Figure 5a) and the set of received signals (Figure 7a) [44]. With respect to the number of sensors in terms of the localization accuracy, previous studies were done to validate the localization algorithm with experimental measurements [38]. A minimum of 4 sensors are needed to obtain reliable results and six sensors have been considered reasonable for this application according to the results of previous studies [38] Thus, it is possible to determine the TDOA and solve the system of equations for the 6 sensor locations tested. The time values have been corrected according to the propagation in the skull that was seen in previous sections. Table 1 shows the positioning of the sensors and the results of the reconstructed position by the location algorithm for energy deposition. The spatial reference for the sensors is given by the software (COMSOL Multiphysics), whereby the position of the source within the skull is known. In case of not considering the different layers, a deviation between the reconstructed position and the simulated source of about 2 mm is obtained. However, in a closer approach, taking into account the different sound speed in the skull with respect to the water, the reconstructed position has been calculated and the results are also shown in Table 1. It can be observed that the deviation is reduced to 1.0 mm or below. 

The calculation time for the reconstruction of the source is approximately 1 ms, thus it can be used for real-time in medical applications on monitor the energy deposition of heavy particle beams.

## 4. Conclusions

Studies have been conducted on the deposition of energy in brain tissue through simulations with Monte Carlo methods for a proton energy beam of 100 MeV. This energy fits the range of clinical applications, taking proton values per pulse, beam width, and time profile that are a compromise between good sound pressure values, valid simulation tools and feasible clinical applications. It has been possible to determine the pressure at a point in space by discretizing the thermoacoustic model and propagating the pressure to a size similar to the Bragg peak. Also, a method of simulation and optimization of the geometry of a piezoelectric ceramic has been applied based on studies on the behaviour of low frequency radial mode. Therefore, through sensors located on the surface of the skull, the position of the source has been reconstructed with 1 mm accuracy from the time of detection by the different sensors considering the corrections in time due to the change of impedance among the propagation media. The position of the source has been solved with few iterations using a robust resolution method for nonlinear equations. To sum up, this technique can be considered a good alternative for monitoring the location of the Bragg peak in the treatment of hadrontherapy, and the simulation study and results presented here can be considered a good starting point before carrying out the experimental evaluation of the technique for clinical purposes.

## Figures and Tables

**Figure 1 sensors-20-02987-f001:**
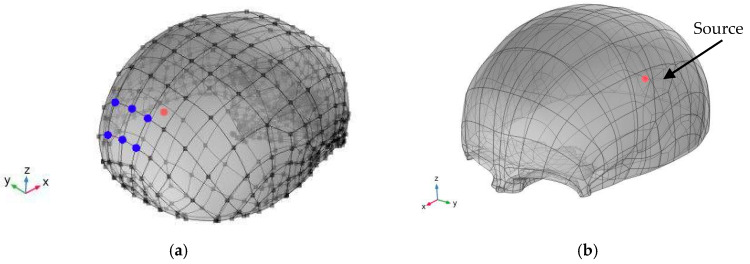
(**a**) Position of the sensors in the skull where the acceleration produced by the propagation of the pressure pulse will be evaluated; (**b**) Simulated volume and pressure source.

**Figure 2 sensors-20-02987-f002:**
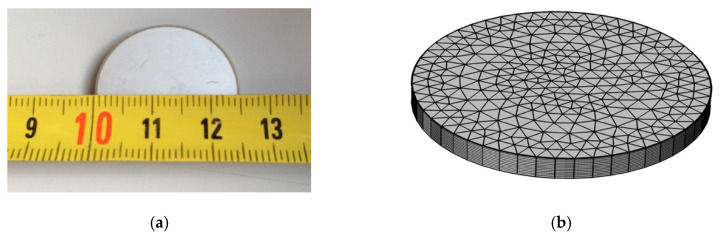
(**a**) PIC 255 Ceramic disc measured; (**b**) Mesh geometry for the finite element method.

**Figure 3 sensors-20-02987-f003:**
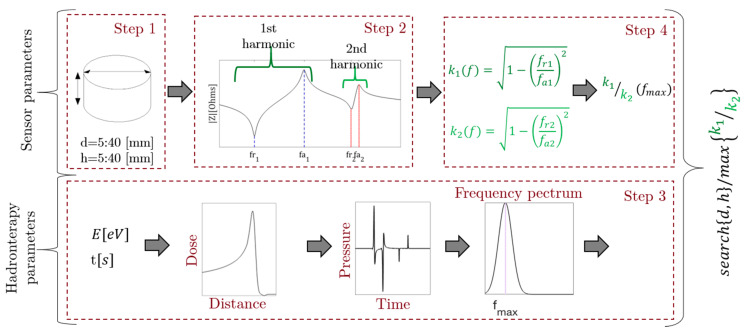
Optimization process diagram with input parameters for the thermoacoustic and piezoelectric parts. Time and energy for the first model and diameter and thickness for second one.

**Figure 4 sensors-20-02987-f004:**
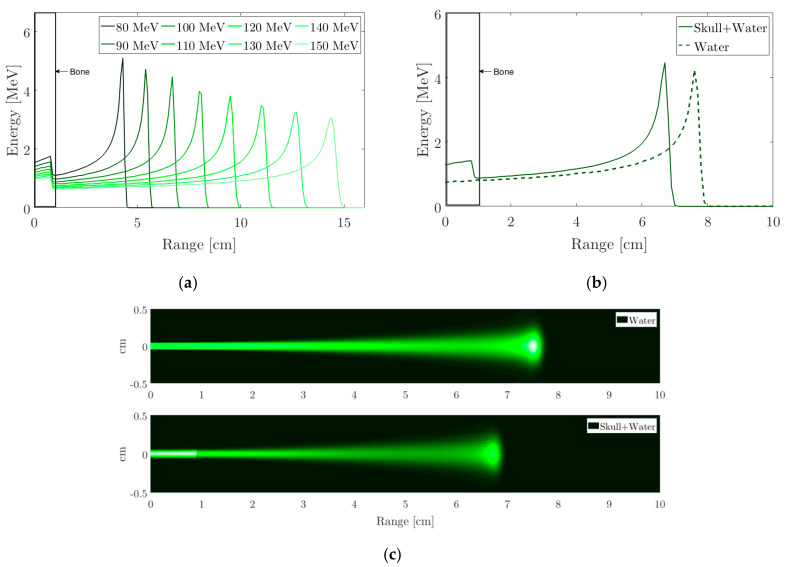
Energy deposition for a proton beam with Gauss profile (σ = 1 mm) and 10^6^ protons. (**a**) layer bone interaction with different beam energies; (**b**) Interaction with the water phantom and with a 1 cm bone layer; (**c**) Deposition in a plane for 100 MeV, the upper figure shows the phantom water and the lower one shows the interaction with a layer of bone.

**Figure 5 sensors-20-02987-f005:**
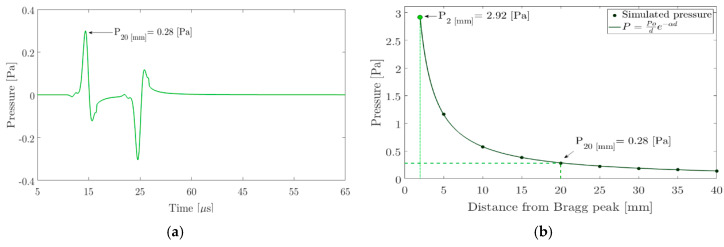
(**a**) Pressure received at 20 mm from the Bragg peak on the proton beam emission axis; (**b**) simulated pressure as a function of the distance.

**Figure 6 sensors-20-02987-f006:**
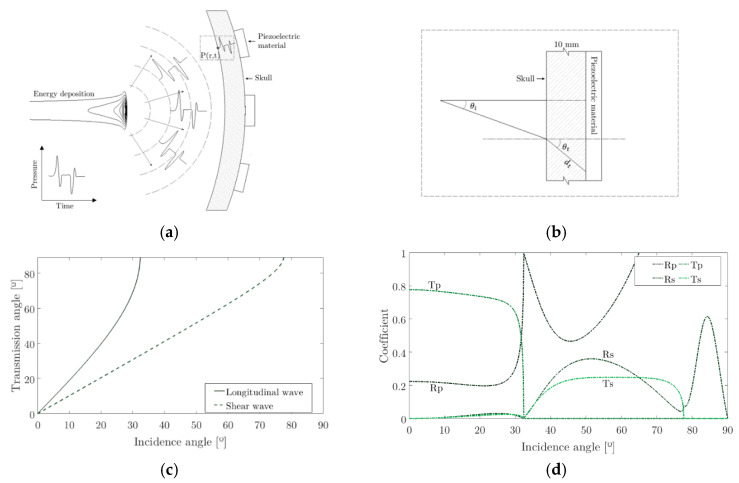
(**a**) Propagation of a longitudinal wave to the point P(r,t) where the speed of sound changes due to the change of medium; (**b**) general diagram of the incidence and transmission angle in the skull; (**c**) transmission angle for longitudinal and shear waves in terms of the incidence angle; (**d**) power transmission and reception coefficients of the cerebrospinal fluid-skull interface.

**Figure 7 sensors-20-02987-f007:**
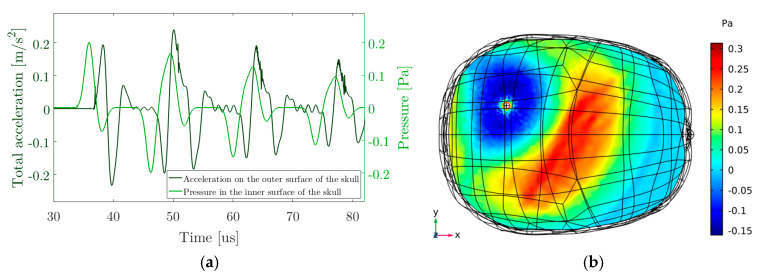
(**a**) Signal received on one of the sensors and pressure on the surface of the skull; (**b**) Propagation in a plane in the X, Y plane 53 μs after the energy deposition.

**Figure 8 sensors-20-02987-f008:**
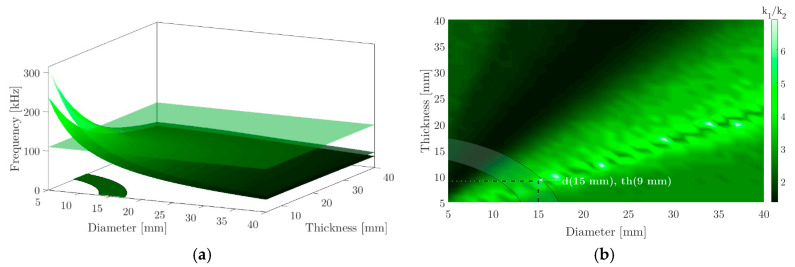
(**a**) Resonance and anti-resonance frequency for the first vibration mode in terms of the diameter and thickness simulated. The horizontal plane represents the central frequency of the pressure pulse of Figure 5a; (**b**) Ratio $k_{1}/k_{2}$ for the first two modes of vibration. The shaded zone represents the area with the best diameter and thickness ratio is optimizing in this region the sensitivity and frequency response of piezoelectric ceramics.

**Figure 9 sensors-20-02987-f009:**
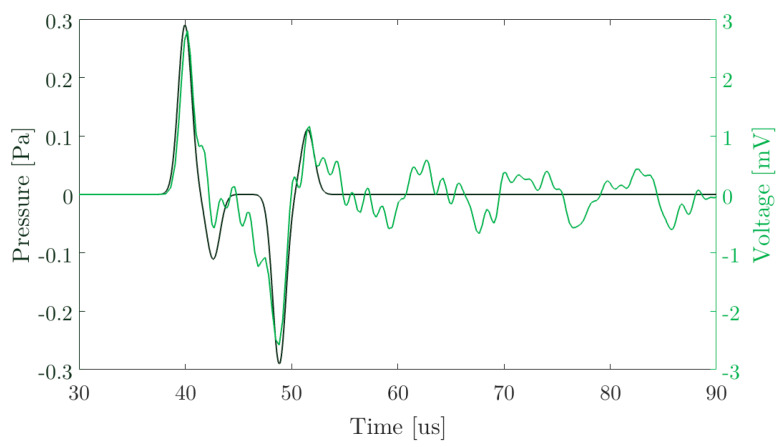
Pressure and terminal voltage for the optimized piezoelectric ceramic for a pressure signal in the emitted source as shown in Figure 5a. The propagated pressure on the sensor surface is 0.28 Pa as shown in Figure 7a.

**Figure 10 sensors-20-02987-f010:**
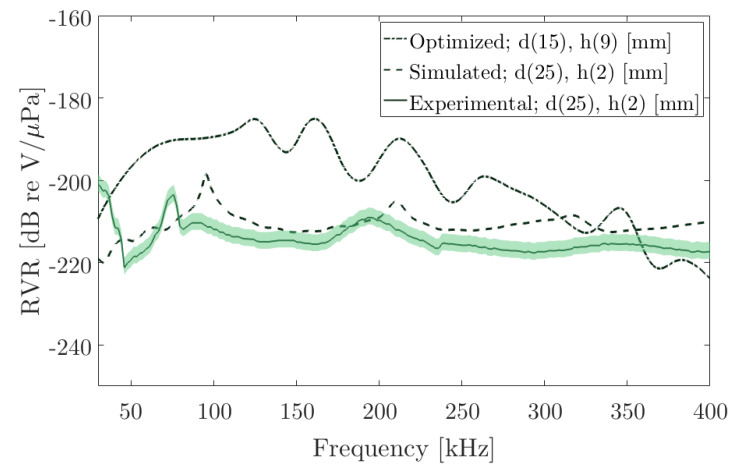
Receiving Voltage Response (RVR) for the piezoelectric ceramic PIC255 with diameter 25 mm and thickness 2 mm measured in the laboratory (solid line) together with its simulation (dotted line), the green area represents the standard deviation in the measurements. The optimized geometry shows a significant increase in the low-frequency band around 110 kHz.

**Table 1 sensors-20-02987-t001:** Estimated position of the Bragg peak energy deposition.

	Sensor [mm]						Source Position [mm]	Reconstructed Position [mm]
	1	2	3	4	5	6		
X	−111.7	−11.1	−10.9	−10.1	−10.1	−100.0	−70.00	−69.60
Y	1.1	13.8	29.8	1.0	15.6	33.5	20.00	20.81
Z	172.2	172.1	171.5	174.0	174.1	173.4	171.90	172.30

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
