# Peer review of "Bragg Peak Localization with Piezoelectric Sensors for Proton Therapy Treatment"

_sensors, 2020, doi:10.3390/s20102987_

Round 1

Reviewer 1 Report

This study conducted a numerical simulation to determine the feasibility and accuracy of acoustic beam monitoring in hadrontherapy. For the localization of the Bragg peak in the treatment of hadron therapy, they utilized the ultrasound piezoelectric sensors at multiple locations in the brain. Authors utilized the typical thermoacoustic models which are similar to photoacoustic models. Also, for the localization of the thermoacoustic signals from the energy source inside the brain, the conventional acoustic signal measurement was used. However, the overall study only showed simulation results by using the typical method for conventional ultrasound beamforming and transducer simulation models. Therefore, I think that the results were not significant. 1. Also, the models, which authors assumed, were too simple. The models are therefore not matched with real clinical models. Typically, the thickness of the skull varies along with the location of the brain. Therefore, for HIFU, MRI or CT imaging are performed to measure the thickness of the skull to focus ultrasound beams inside the brain with high accuracy. 2. Why did not authors use the KLM model for the piezoelectric sensors? The KLM model is a well-known model for ultrasound transducer. 3. Also, the sound speed is dependent on the frequency of ultrasound. The authors did not consider these points to find the accurate location of the Bragg peaks. The authors should consider this. 4. For the accurate simulation of acoustic propagation, authors should consider the acoustic properties of different layers of the brain. Why don't you consider the acoustic properties of the internal structures of the brain? 5. To make this study significant, the experimental evaluation for the models should be required.

Author Response

Find attached the comments to reviewer 1.

Reviewer 2 Report

In this manuscript, the authors proposed a full chain simulation of the acoustic hadrontherapy monitoring for brain tumours. Geant4 is used to simulate the energy deposition and to study the behaviour of the Bragg peak.  The resulting thermoacoustic signal has been subsequently obtained by solving the thermoacoustic equation. The acoustic propagation has been simulated by FEM methods in the brain and the skull. Localizing Bragg peak based on thermoacoustics is interesting and attractive. However, I have a few concerns below.

  1. Proton induced acoustics has been simulated intensively in recent years. However, no literatures have been reviewed in the introduction section. Please review the recent progress in this section.
  2. In the simulation, the proton beam time profile is defined as 10 microseconds and a total of 5 million protons per pulse. It’s unclear these parameters are available in clinic.
  3. It’s unclear what the central frequency and bandwidth of acoustic transducer being used in the simulation.
  4. How many percentages of the thermoacoustic signal will be received by the acoustic transducer after the attenuation and reflection from the skull?
  5. How many acoustic transducers are needed for localizing the protoacoustic signal source?

Author Response

Find attached the comments to reviewer 2.

Round 2

Reviewer 1 Report

Although the authors addressed the raised concerns, the authors should clearly demonstrate that the simulation study shows a new finding. The most of simulation results have been extensively studied in ultrasound and photoacoustic imaging fields.

Also, what is the target resolution to be monitored? Authors should determine whether the sound speed variation is important to monitor the hydron therapy as the target resolution. In HIFU therapy, the compensation of sound speed in the skull is very important for precise therapy of the brain since the mistreatment of the brain would result in crucial damages to patients. 

Also, the ultrasound frequency changes the natural focal point up to 5mm from 1 - 5MHz, which depending on tissues. 

Author Response

Find attached the answers.

Round 3

Reviewer 1 Report

No further comments